# Seasonal Occurrence and Relative Abundance of Marine Fish Larval Families over Healthy and Degraded Seagrass Beds in Coastal Kenya

**James M. Mwaluma** [1,*], **Gladys M. Okemwa** [1], **Alphine M. Mboga** [2], **Noah Ngisiange** [1], **Monika Winder** [3], **Margareth S. Kyewalyanga** [4], **Joseph Kilonzo** [1] **and Immaculate M. Kinyua** [1]

[1]   Kenya Marine and Fisheries Research Institute (KMFRI), Mombasa P.O. Box 81651-80100, Kenya
[2]   School of Environment and Earth Science, Department of Environmental Science, Pwani University, Kilifi P.O. Box 195-80108, Kenya
[3]   Department of Ecology, Environment and Plant Sciences, Stockholm University, SE-106 91 Stockholm, Sweden
[4]   Institute of Marine Sciences (IMS), University of Dar es Salaam, Zanzibar P.O. Box 668, Tanzania
[*]   Correspondence: jamesmwaluma@gmail.com

**Abstract:** Seagrass beds provide critical nursery habitats and spawning grounds for new generations of fish. The habitats are under threat from human activities and climate change, and with that, an important ocean service is lost that limits fish production. The present study investigates patterns in the larval occurrence and abundance in seagrass meadows at two locations with varying degrees of seagrass fragmentation. Monthly ichthyoplankton sampling was conducted during the northeast monsoon (NEM) and southeast monsoon (SEM) seasons in 2019 and 2020. A total of 42 larval fish families belonging to 37 genera and 21 species were identified. Dominant families were Labridae (29.5%), Blenniidae (28.7%), Gobiidae (26.0%), Engraulidae (23.3%) and Scaridae (22.3%). Canonical Correspondence Analysis and regression analysis revealed water temperature, dissolved oxygen and pH as the most important abiotic variables driving taxonomic composition of larval assemblages, while zooplankton and chlorophyll-*a* were the most important biotic factors. Fish larvae were more abundant in healthy seagrass habitats as compared to degraded ones. However, despite some loss in functionality, the degraded sites equally played a role in supporting some species, including Gobiidae and Blenniidae. Seasonality influenced larval abundance at the two sites, with a peak in mean abundance coinciding with the NEM season. Interannual variability in fish larval abundance was observed at both sites, indicating that factors controlling larval production varied between the years. This study demonstrates the important role of seagrass meadows in the replenishment of fish stocks and supportive evidence for their management and conservation.

**Keywords:** fish larvae; abundance; seasonality; seagrass habitats; Kenya

## 1. Introduction

Seagrass beds provide critical settlement and nursery habitats for fish [1–3]. Their ecological health is, therefore, a vital factor in maintaining high survival rates of larval and juvenile fish. However, these essential nursery habitats are under threat due to human activities and climate change [4]. Recruitment of reef fish species specifically depends on shallow coastal habitats including seagrass beds, which provide settlement and nursery habitats [1–3]. However, these habitats continue to experience degradation resulting in fragmentation. Overfishing and use of destructive fishing gears such as beach seines are primary threats to the ecological integrity and functioning of seagrass meadows, further contributing to a decline in fishery resources, particularly within the Western Indian Ocean (WIO) region [5–7]. Various studies have shown that patchy seagrass meadows have lower species richness and diversity when compared to continuous meadows, thus limiting their nursery function [6,8]. Conservation of these near-shore habitats to enhance fish

recruitment has, thus, become an important part of fishery management [9]. However, a key question is how increasing degradation of seagrass beds limits the production and dispersal of fish larvae, a critical bottleneck for sustainable fish stocks [10].

Studies conducted on fish larvae taxonomy, dispersal, distribution, and abundance in coastal and estuarine waters of the WIO region include Kaunda-Arara et al. [11], Mwaluma et al. [12–14] and Crochelet et al. [4,15]. In this study, we describe seasonal patterns of fish larval occurrence and abundance in seagrass habitats with varying degrees of fragmentation and relate this to biophysical factors and seasonal cycles. A better understanding of how degradation of seagrass meadows affects fish recruitment, distribution and diversity will provide supportive evidence for their management and conservation.

## 2. Materials and Methods

This study focused on two sites, Watamu, located at the north coast of Kenya, and Diani, located at the south coast (Figure 1).

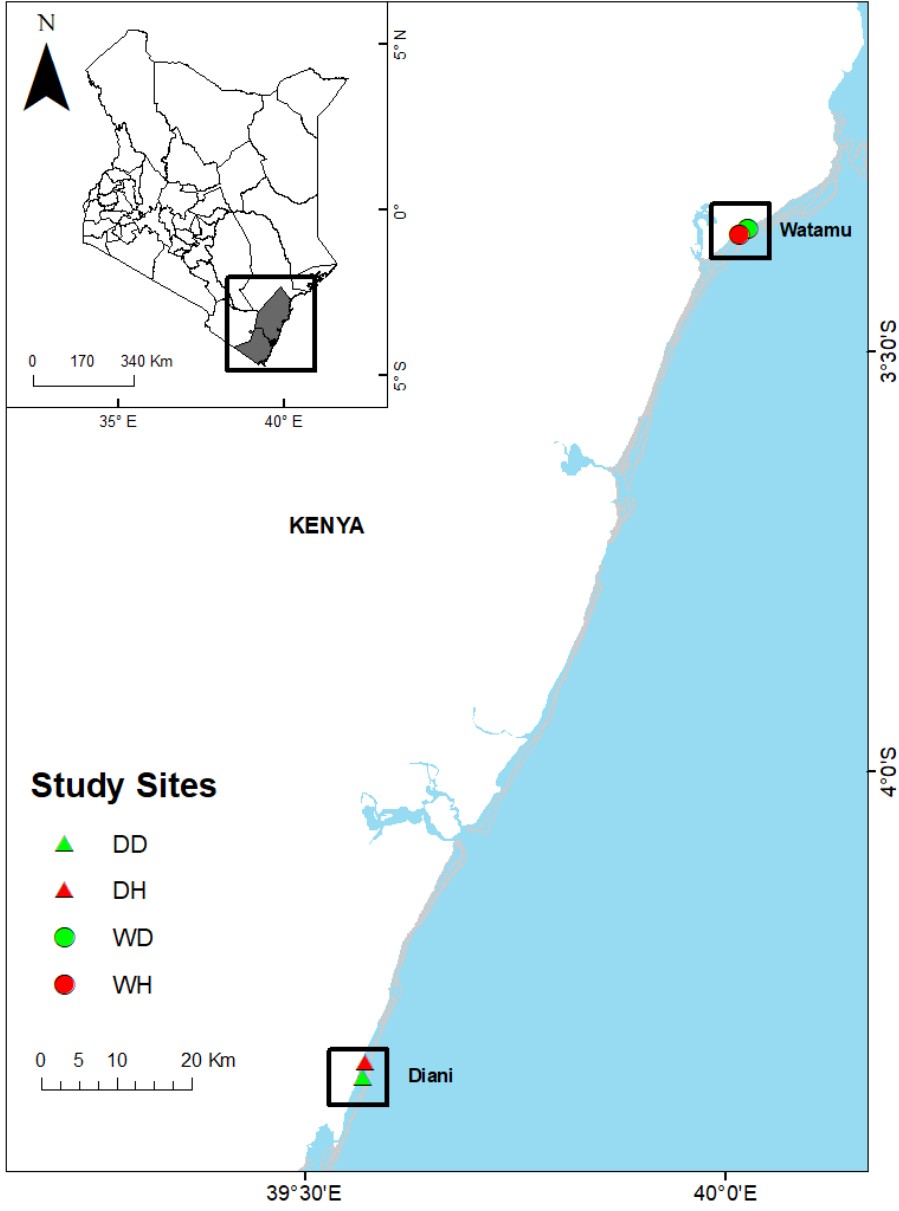

**Figure 1.** Map showing the location of the study sites and stations in Watamu and Diani in coastal Kenya.

Both sites are shallow lagoons characterized by a mixed semi-diurnal tide with two maxima and two minima per day with a tidal range of about 2.0 m at the neap tide and 2.9 m during the spring tide. Sampling at each site was conducted along transects lying 3 km from shore at two stations, one representing a healthy site and the other representing a degraded site. "Healthy sites" comprised relatively continuous seagrass cover composed of 65% and above, while "degraded sites" comprised fragmented seagrass zones with a cover of less than 65% interspersed with sandy bottoms. Watamu stations were coded as Watamu Healthy (WH) and Watamu Degraded (WD), while in Diani, the stations were coded as Diani Healthy (DH) and Diani Degraded (DD). DH was dominated by *Thalasodenron ciliatum* seagrass cover, while DD was located 2 km apart with a discontinuous seagrass zone of *T. hemprichii* and *T. ciliatum* interspersed with sandy bottoms. Watamu stations consisted of mixed meadows comprising of pioneer species, initial stages of recovery from what may have been as a result of previously reported urchin herbivory that had severely affected the region, while Diani stations were dominated by climax communities of *T. ciliatum*. Seagrass degradation at both study sites was first reported by Zanre [16] and Uku [17], evidenced by an extensive proliferation of the sea urchin *Tripneustes gratilla*. Recovery at the sites has since been slow due to overfishing of *T. gracilis* predators in several parts of the coast [18]. Other likely causes of degradation at the sites include use of destructive fishing gears (beach seines), collection of seagrass for use as bait and climate change [19].

### 2.1. Environmental Variables

Dissolved oxygen, chlorophyll-*a*, temperature, salinity and pH were measured monthly at each station using a YSI Multiparameter probe. Water samples for chlorophyll-*a* (Chl-*a*) were collected at each station. Chl-*a* samples (surface water) were collected using 1.5 L plastic bottles at each site and then filtered through 0.45 μm millipore membrane filter for laboratory analysis.

### 2.2. Plankton Sampling

At each station, replicate surface water plankton tows were made by towing horizontally a bongo net of 0.5 m diameter fitted with 500 μm mesh size net fastened with a General Oceanics flowmeter for 20 min. Towing was conducted between 09:00 h and 14:00 h against the incoming tide at a constant speed of between 0.5–1.8 knots. Collected samples were labeled and preserved using 5% formaldehyde buffered in seawater. Sampling was carried out in June, July and August in 2019 and 2020 to represent the SEM seasons, while sampling during the NEM season was conducted in November and December 2019 and 2020 and January 2020 and 2021.

In the laboratory, fish larvae were sorted from the zooplankton samples and separated in vials filled with 90% alcohol. Fish larvae were later removed from the vials, sorted into families and identified to the nearest species using identification keys [20–23] and counted. The rest of the zooplankton were identified, subsampled and counted.

Counting of zooplankton was performed in graduated petri dishes and the totals were multiplied by the dilution factor (250 mL/20 mL) to derive the total abundance estimate. This estimate was subsequently divided by the total volume of seawater filtered during the net haul (m$^3$) with final zooplankton densities expressed as no·m$^{-3}$. The keys and identification references used were obtained from [24–33] Density for both fish larvae and zooplankton was expressed as no·100 m$^{-3}$ and no·m$^{-3}$, respectively.

Fish eggs were then separated from the same sample, counted whole and expressed as numbers per cubic meter (no·m$^{-3}$). No attempt was made at assigning them to species due to the difficulty in identification and limited guides.

### 2.3. Data Analysis

Plankton abundance was log (x + 1) transformed to fulfill the normality requirements for parametric statistical analysis. Monthly data were considered replicates in each season. Analysis of variance (ANOVA) was then applied to test for differences in fish larval

densities, zooplankton, fish eggs and chlorophyll-*a*, pH, DO, salinity and temperature in different locations (Watamu vs. Diani), seasons (NEM vs. SEM) and sites (Healthy vs. Degraded). Canonical Correspondence Analysis (CCA) was performed using PAST ver. 4.1.1 software to visualize the association between environmental variables and dominant fish larval families. Regression analysis was done to investigate the influence of biotic factors (Chl-*a*, zooplankton density and fish egg abundance) on overall fish larvae supply. Other biological factors that influence larval supply, such as phytoplankton density, were not considered in this study.

## 3. Results

### 3.1. Environmental Variables

Dissolved oxygen exhibited a seasonal pattern with higher values experienced during the SEM compared to NEM season. Higher values were obtained from Watamu compared to Diani (Figure 2A).

Chlorophyll-*a* exhibited a similar pattern to dissolved oxygen with higher values recorded during the SEM compared to the NEM. Higher values were obtained from Watamu site compared to Diani (Figure 2B) in 2019 during both seasons. Chlorophyll-*a* was highest in 2019 compared to 2020 and 2021, indicating interannual variability and implying that factors that were affecting productivity could have been different over the years. No significant differences were observed between stations in Diani and Watamu (Table S1).

Mean temperature and salinity varied between seasons, with lower salinities and temperatures recorded during the SEM. The highest temperature was recorded in November in Diani during the NEM 2020, while the lowest was in August the same year (Figure 2C). The highest salinity was observed in Watamu in November 2020, while the lowest was recorded in the same site (Figure 2D).

The mean pH during NEM was $8.08 \pm 0.02$ and ranged from 7.85–8.28. During SEM, the mean pH was $8.20 \pm 0.03$ and ranged between 7.99–8.54 (Figure 2E). Mean pH values differed significantly between seasons ($p = 0.003$), but did not differ significantly between sites (Table S1).

### 3.2. Zooplankton, Fish Eggs and Larval Fish Abundance

Zooplankton varied seasonally between the healthy and degraded sites (Figure 3A). Overall zooplankton density was highest during the NEM season, with a peak of 301 no·m$^{-3}$ in the degraded site at Diani while the lowest was 24 no·m$^{-3}$ during the SEM season also in the degraded site at Diani (Figure 3A). The healthy and degraded sites did not differ significantly despite higher values in Watamu; however, significant differences were recorded between seasons ($p = 0.008$). Zooplankton abundance in Diani was more variable in general, whereas Watamu was more consistent, but the pattern of degraded vs. healthy was not discernible at either site.

### 3.3. Fish Larvae

The highest fish larval abundance was 55 no·100 m$^{-3}$ recorded in 2019 during the NEM season in Watamu, while the lowest was 2.0 no·100 m$^{-3}$ during the same season in 2020 in Watamu (Figure 3C). Overall, healthy sites had a higher abundance of fish larvae compared to degraded sites; however, the difference was not significant ($p = 0.109$).

### 3.3.1. Species Composition and Abundance

A total of 42 families of fish larvae belonging to 37 genera and 21 species were identified. Dominant fish families were Labridae, Engraulidae, Blenniidae, Scaridae and Sphyraenidae. These families were predominant during the NEM in both sites (Table 1).

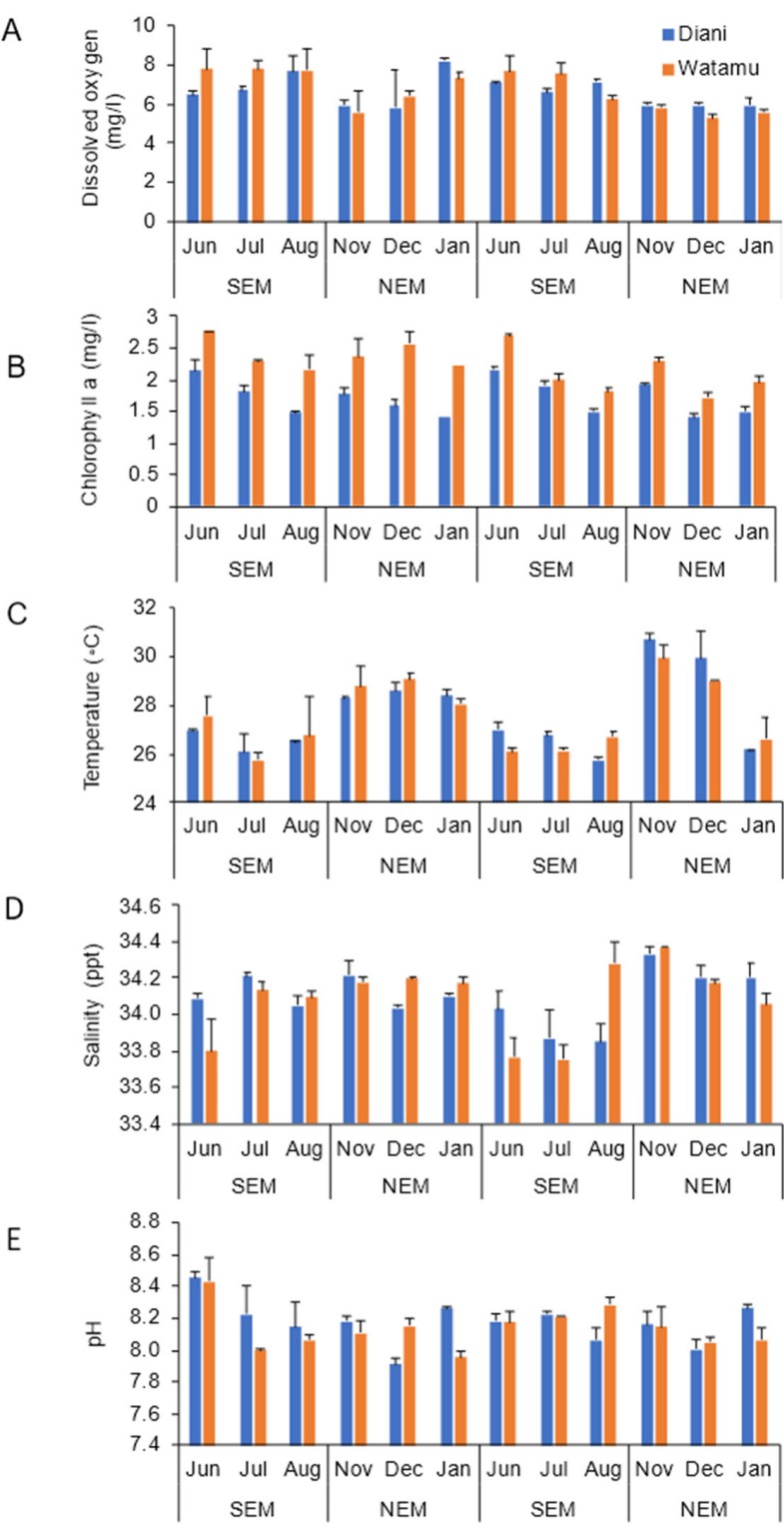

**Figure 2.** (**A–E**) Variation in dissolved oxygen (**A**), chlorophyll-*a* (**B**), temperature (**C**), salinity (**D**) and pH (**E**) in Watamu and Diani, Kenya (±SE) during June to December 2019, January to December 2020 and January 2021.

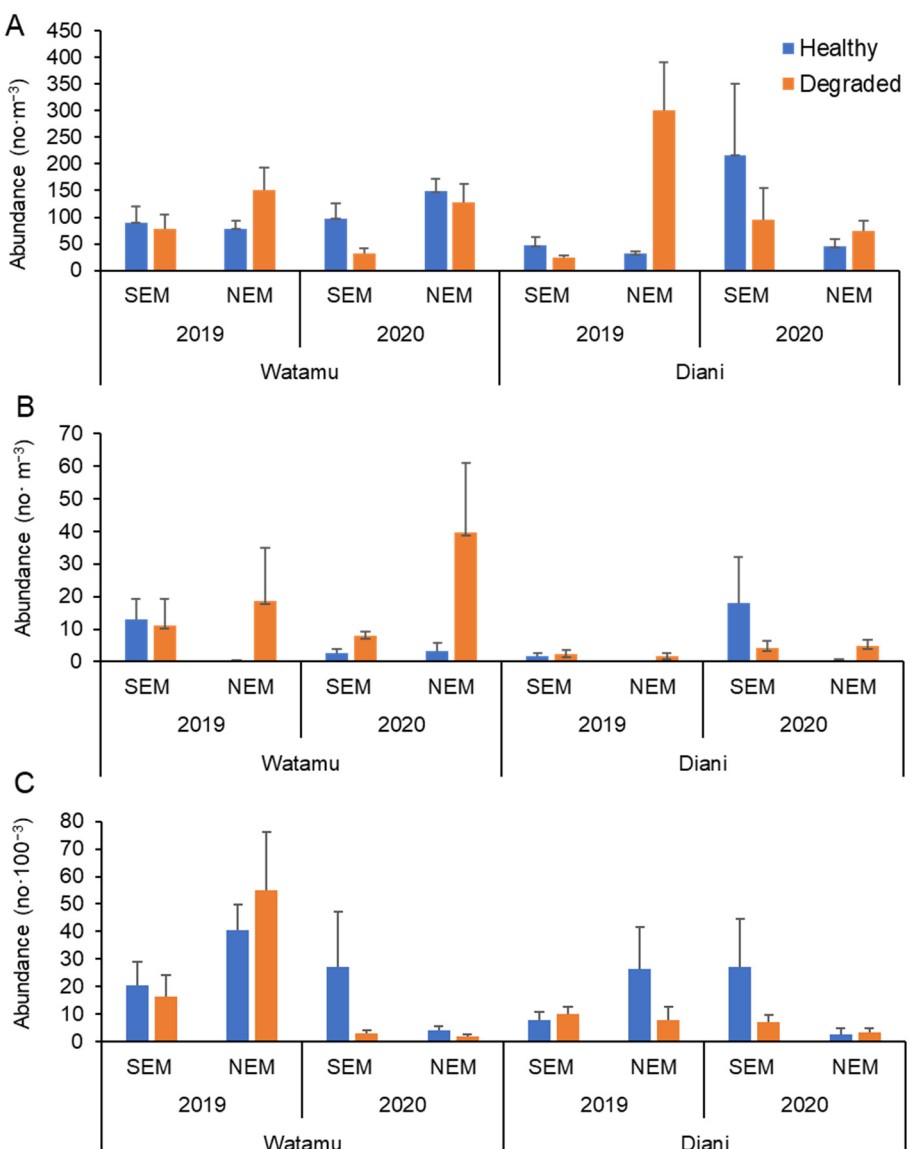

**Figure 3.** Seasonal variation in (**A**) zooplankton, (**B**) fish eggs and (**C**) fish larvae abundance in Watamu and Diani, Kenya in the years 2019–2020 ($\pm$SE). Fish eggs followed a similar pattern with zooplankton density. The highest fish egg densities recorded was 40 no·m$^{-3}$ during the NEM season Watamu, while the lowest was 0.3 no·m$^{-3}$ at the same site (Figure 3B). Fish egg abundance in Watamu was higher than Diani but not statistically significant ($p$ = 0.06).

**Table 1.** Total larval abundance (no·100$^{-3}$) during the northeast and southeast monsoon season and overall percentage contribution in Diani and Watamu.

| Sites | Diani | | | Watamu | | |
|---|---|---|---|---|---|---|
| Family | NEM | SEM | Overall (%) | NEM | SEM | Overall (%) |
| Acanthuridae | 1.0 | 0.0 | 0.1 | 2.0 | 0.0 | 0.3 |
| Atherinidae | 0.0 | 0.0 | 0.0 | 2.6 | 0.0 | 0.4 |
| Apogonidae | 19.2 | 2.5 | 3.1 | 16.2 | 5.8 | 3.3 |
| Atherinidae | 15.4 | 3.4 | 2.7 | 9.2 | 0.0 | 1.4 |
| Belonidae | 0.0 | 0.6 | 0.1 | 0.0 | 0.0 | 0.0 |
| Blenniidae | 124.8 | 4.2 | 18.5 | 55.4 | 12.1 | 10.2 |
| Bothidae | 16.2 | 0.0 | 2.3 | 0.0 | 0.0 | 0.0 |
| Callionymidae | 0.0 | 0.0 | 0.0 | 1.9 | 0.0 | 0.3 |

**Table 1.** *Cont.*

| Sites | Diani | | | Watamu | | |
|---|---|---|---|---|---|---|
| **Family** | **NEM** | **SEM** | **Overall (%)** | **NEM** | **SEM** | **Overall (%)** |
| Carangidae | 1.9 | 3.4 | 0.8 | 0.0 | 0.0 | 0.0 |
| Diodontidae | 8.1 | 0.0 | 1.2 | 0.9 | 0.0 | 0.1 |
| Eleotrididae | 1.0 | 0.0 | 0.1 | 0.0 | 0.0 | 0.0 |
| Engraulidae | 136.3 | 4.6 | 20.2 | 18.4 | 1.4 | 3.0 |
| Exocoetidae | 0.0 | 0.0 | 0.0 | 8.1 | 0.0 | 1.2 |
| Gerreidae | 0.0 | 1.3 | 0.2 | 1.5 | 1.1 | 0.4 |
| Gobiidae | 28.4 | 8.3 | 5.3 | 117.5 | 19.2 | 20.7 |
| Haemulidae | 0.0 | 1.1 | 0.2 | 0.0 | 0.0 | 0.0 |
| Hemiramphidae | 8.1 | 2.7 | 1.5 | 0.0 | 0.0 | 0.0 |
| Istiophoridae | 0.0 | 0.0 | 0.0 | 16.2 | 0.0 | 2.4 |
| Labridae | 141.6 | 0.5 | 20.4 | 51.2 | 9.2 | 9.1 |
| Leiognathidae | 1.1 | 0.0 | 0.2 | 1.1 | 0.0 | 0.2 |
| Lethrinidae | 0.7 | 0.5 | 0.2 | 1.1 | 0.0 | 0.2 |
| Lutjanidae | 4.4 | 1.9 | 0.9 | 2.8 | 0.0 | 0.4 |
| Monodactylidae | 0.0 | 0.0 | 0.0 | 8.1 | 5.0 | 2.0 |
| Mullidae | 2.5 | 0.0 | 0.4 | 0.0 | 0.0 | 0.0 |
| Myctophidae | 0.0 | 0.0 | 0.0 | 64.8 | 0.0 | 9.8 |
| Nemipteridae | 9.1 | 0.0 | 1.3 | 50.7 | 3.1 | 8.1 |
| Nomeidae | 0.0 | 0.0 | 0.0 | 8.1 | 0.5 | 1.3 |
| Ostraciidae | 1.6 | 0.0 | 0.2 | 0.0 | 7.8 | 1.2 |
| Platycephalidae | 1.8 | 3.9 | 0.8 | 8.1 | 4.2 | 1.9 |
| Pleuronectidae | 0.0 | 0.5 | 0.1 | 0.0 | 0.0 | 0.0 |
| Pomacentridae | 4.8 | 2.2 | 1.0 | 3.0 | 0.0 | 0.5 |
| Scaridae | 77.0 | 3.6 | 11.6 | 49.4 | 21.1 | 10.6 |
| Scombridae | 0.0 | 0.5 | 0.1 | 0.0 | 0.0 | 0.0 |
| Scorpaenidae | 0.0 | 0.5 | 0.1 | 1.1 | 7.5 | 1.3 |
| Serranidae | 0.0 | 0.9 | 0.1 | 16.2 | 1.0 | 2.6 |
| Sillaginidae | 0.0 | 0.5 | 0.1 | 0.0 | 0.0 | 0.0 |
| Solenostomidae | 7.4 | 0.8 | 1.2 | 2.9 | 8.0 | 1.6 |
| Sphyraenidae | 16.2 | 0.0 | 2.3 | 0.9 | 0.0 | 0.1 |
| Syngnathidae | 9.1 | 5.8 | 2.1 | 34.7 | 0.9 | 5.4 |
| Terapontidae | 1.3 | 0.0 | 0.2 | 0.0 | 0.0 | 0.0 |
| Tetraodontidae | 0.7 | 2.6 | 0.5 | 0.0 | 0.0 | 0.0 |
| Trichonotidae | 0.7 | 0.0 | 0.1 | 0.0 | 0.0 | 0.0 |
| Grand Total | 640.4 | 56.8 | | 554.1 | 107.9 | |

### 3.3.2. Distribution of Larval Fish Families

At both study sites, larval fish families were predominantly abundant in seagrass healthy sites comprising of the families Blenniidae, Engraulidae, Labridae, Scaridae, Myctophidae, Nemipteridae, Sphyraenidae and Atherinidae (Figure 4). Degraded sites were dominated by Gobiidae, Syngnathidae, Serranidae, Platycephalidae and Solenostomatidae. Lethrinidae, Lutjanidae and Scorpaenidae larvae were generally scarce at both study sites.

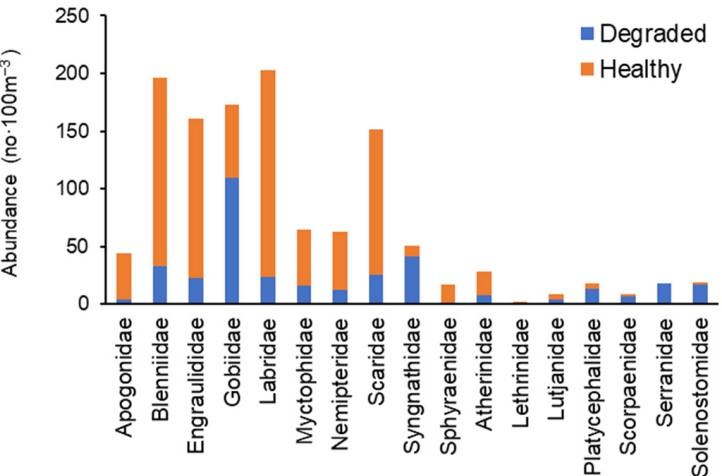

**Figure 4.** Composition and abundance of larval fish families in degraded and healthy seagrass habitats.

### 3.4. Influence of Biophysical Variables and Seasonality

Among the environmental variables measured, the CCA biplot of the 12 dominant families revealed sea water temperature as the most important variable driving taxonomic abundance of fish larvae in the study sites (Figure 5). The abundance of Nemipteridae and Blenniidae was negatively associated with decreasing salinity levels (Figure 5).

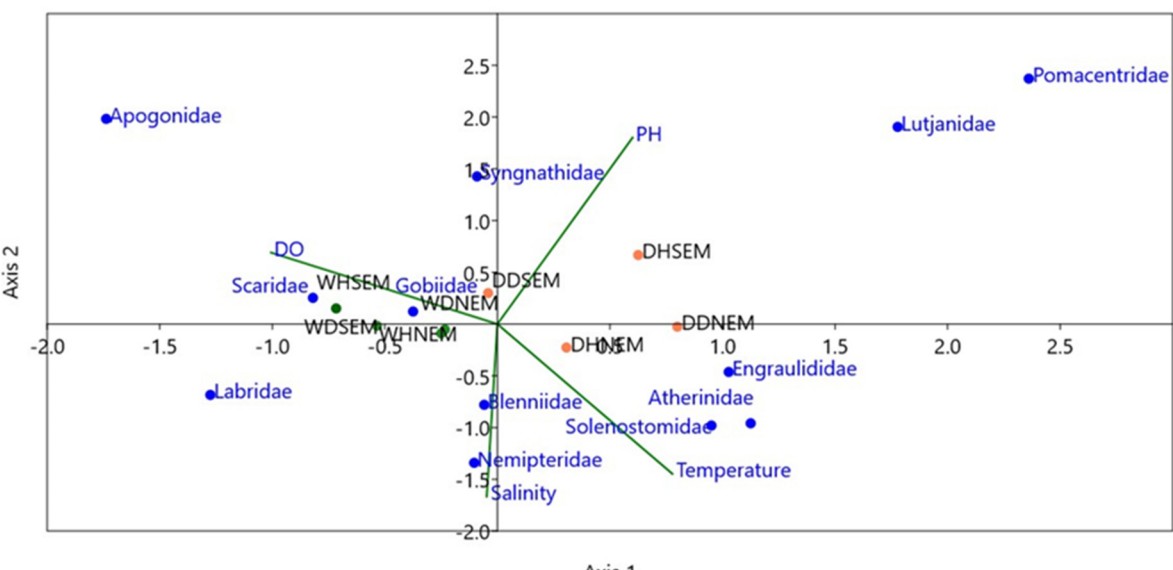

**Figure 5.** Canonical correspondence analysis (CCA) biplot depicting association of environmental variables (dissolved oxygen, salinity and temperature) among species at the study sites (Diani vs. Watamu), as well as status (Healthy vs. Degraded) with environmental variables presented as vectors (DD = Diani degraded, DH = Diani healthy, WD = Watamu degraded, WH = Watamu healthy).

In Diani, the CCA further revealed seasonality as key driver of fish larval assemblage structure. Engraulidae, Solenostomidae and Atheridinae larvae were strongly associated with increasing temperature but negatively associated with decreasing salinity indicating that they were least abundant when salinity was low. Pomacentridae and Lutjanidae larvae were negatively associated with increasing temperature. In Watamu, increasing levels of dissolved oxygen strongly influenced the abundance of Apogonidae, Scaridae and Gobiidae larvae. Nemipteridae and Blenniidae larvae negatively correlated with increasing salinity levels.

The results of the regression analysis revealed that zooplankton and Chl-*a* had a strong positive correlation with fish larval abundance (Table 2).

**Table 2.** Regression summary of the correlation between fish larvae density and biotic factors (Phy = Phytoplankton, Zoop = Zooplankton, abn = abundance, S.E. = Standard Error).

| *n* = 56 | Beta | S.E. | B | S.E. | T (51) | *p*-Level |
|---|---|---|---|---|---|---|
| Intercept | | | 0.2525 | 0.501 | 0.5031 | 0.617 |
| Chl-*a* | 0.571 | 0.0998 | 1.529 | 0.267 | 5.718 | * 0.000 |
| Phy abn | −0.131 | 0.0999 | −0.210 | 0.159 | −1.314 | 0.194 |
| Zoop abn | 0.601 | 0.1242 | 0.687 | 0.141 | 4.844 | * 0.000 |
| Fish eggs | −0.077 | 0.0122 | −0.063 | 0.101 | −0.632 | 0.529 |

* Significant at *p* = 0.05.

## 4. Discussion

The ichthyoplankton assemblages reported in this study is comparable to previous findings of other studies along the Kenyan coast [12–14]. Over 60% of the assemblage

was dominated by Blenniidae, Engraulidae, Gobiidae, Labridae and Scaridae larvae, suggesting that seagrass habitats play a major role in supporting these families. Bleniidae and Gobiidae, which are demersal spawners, were the most abundant, while larvae of the pelagic spawners Lethrinidae, Lutjanidae and Siganidae were particularly scanty. It is important to acknowledge that small, benthic/demersal spawners such as Blenniidae and Gobiidae do not undergo mass migrations to offshore or pelagic spawning grounds, and are, thus, likely to spawn locally (self-recruit), and hence, be well represented in plankton samples [12]. Notably, pelagic spawners are likely to migrate to deeper offshore coastal waters to spawn, and hence, likely to be poorly represented in plankton samples in and around shallow-water seagrass beds [34,35]. Thus, the composition of fish larval families reported in this study is strongly influenced by the spawning strategies of the species and the sampling method used.

The high numbers of larval fish In Watamu compared to Diani was likely due to the higher zooplankton density in Watamu and a diverse network of pioneer seagrass communities, which created suitable habitats for the larvae [18,36]. Additionally, Watamu is located adjacent to a Marine Park and could be experiencing larval seeding from the park [13]. However, the two study sites are not comparable on at least two scores: firstly, the geographic separation and, secondly, the different types of seagrass species [37]. The higher abundance of fish larvae during the NEM season in both sites suggests a typical phenomenon for the Kenyan coast area as observed by Mwaluma et al. 2011 [13]. The general peak in fish larval abundance is likely correlated with a peak in spawning during the NEM. The higher temperature and stable water conditions that prevail during the NEM season trigger optimum conditions, which increase phytoplankton and zooplankton productivity along the Kenya coast [34–36]. However, it is important to note that seasonal patterns of larval abundance will be species-specific, with some species peaking during the SEM season [13,38].

This study reported interannual variability in fish larval abundance at both study sites, indicating that factors controlling larval production vary between years and are influenced by seasonal dynamics, evidenced by changes in species composition [11], with distinct taxa and groups showing intra-annual peaks, a common feature in the tropics. Early studies conducted along the coast of Kenya found interannual variability in the species composition and abundance of fish larvae to be associated with lunar and seasonal changes in environmental processes [11–14]. More recent studies have further attributed variations in fish larval abundance to variations in fish spawning patterns, variations in larval supply and retention, zooplankton abundance and seasonality [4,39,40].

The survival and recruitment success of fish larvae largely depends on a coincidence of favorable abiotic and biotic factors [40]. Temperature was the primary abiotic driver of fish larval abundance in this study, while zooplankton and Chl-*a* were the important biotic factors, indicating the timing of fish reproduction to suitable biophysical conditions [11,13,39]. The lower pH values during SEM could be due to the influence of surface run-off of rainwater into the sea. Although pH was not a primary influential factor, studies have shown that lower pH values may affect fish larvae growth rates [41], and hence, should not be ignored.

The significantly higher abundance of fish eggs encountered in Watamu degraded site during the NEM was most likely due to an abundance of Blenniidae and Gobiidae observed in the larval samples. These species are known to be self-recruiting demersal spawners with a preference for sandy open areas [13]. Eggs of the demersal spawning Syngnathidae, Platycephalidae and Serranidae were also associated with open sandy areas, while eggs of the demersal spawning Blenniidae, Labridae and Scaridae eggs were highly abundant in seagrass sites. Most interesting was that eggs of Engraulidae, which are pelagic coastal water spawners, were also well represented in seagrass bed samples. Being planktivorus carnivores, Engraulidae are highly likely to be attracted to chemical cues from planktonic phases of crustaceans, gastropods and bivalves within the seagrass beds. Other factors reported to influence fish larval habitat associations and distribution of fish eggs to seagrass sites include adult spawning behavior [42,43], larval settlement behavior [44], physical

and oceanographic processes (tides, currents) [45] and lunar cycles [13]. It has also been postulated that dense seagrass cover and plant detritus are correlated with zooplankton and phytoplankton abundance which constitute a major food source for ichthyoplankton and mesopredators, including fish [13,46].

## 5. Conclusions

This study investigated seasonal abundance patterns of fish larvae in seagrass meadows at two sites of varying fragmentation levels along the Kenya coast. Results showed differences in species abundance of fish larvae between healthy and degraded sites and between seasons. Overall, healthy and degraded sites did not differ significantly in the biotic factors; however, seasonal differences were observed, with the most important abiotic factors being temperature and salinity.

A limitation of this study is that plankton tows mainly capture pre-settlement larvae (planktonic), which may influence the results by underestimating other development stages of larvae, particularly those with strong swimming abilities which tend to avoid nets. Light traps would be useful for collection post-settlement larvae, although with biases towards phototactic larvae. Future studies can, therefore, be designed to utilize both methods in order to get a broader representation of taxa and families that settle within each habitat. Additionally, the assessments should focus on validating distribution and seasonal patterns along a perpendicular sampling regime from the lagoon to offshore waters, and also consider the influence of other environmental factors such as lunar cycles.

The study builds on the substantive evidence supporting the important nursery function of seagrass beds reported by Lefcheck et al. [47]. The findings have several implications in conservation and management. Firstly, the study has provided an account of key commercial fish families, utilizing coastal habitats as nursery habitats and the abiotic factors affecting their distribution. Secondly, it has further shown that despite some loss in functionality, degraded sites equally play a role in supporting replenishment of fish in nearshore coastal areas. This may be an important consideration in locating the boundaries of marine parks and conservation areas.

**Supplementary Materials:** The following supporting information can be downloaded at: https://www.mdpi.com/article/10.3390/d14090730/s1, Table S1: Results of the Analysis of Variance testing for differences in fish larval densities, zooplankton, fish eggs and chlorophyll-a, pH, DO, salinity and temperature in different locations (Watamu vs. Diani), seasons (NEM vs. SEM) and sites (Healthy vs. Degraded). Bold *p*-values indicate a significant result of $p < 0.05$.

**Author Contributions:** J.M.M. is the lead author and responsible for coordination of all contributions; G.M.O. contributed to the analysis and interpretation of the data and manuscript review; A.M.M. contributed to the literature review, sample collection and laboratory identifications of fish larvae; N.N. contributed to the generation of the maps and data analysis; M.W. contributed to the analysis and interpretation of the data with contributions from M.S.K. and G.M.O.; M.W. and M.S.K. further contributed to proof reading and improvement of the manuscript. J.K. and I.M.K. were involved in sample collection and laboratory analysis. All authors have read and agreed to the published version of the manuscript.

**Funding:** This research was funded by the Western Indian Ocean Marine Science Association (WIOMSA) grant no: MASMA/OP/2018/01.

**Acknowledgments:** We wish to thank the technical team from KMFRI for their assistance and dedication to sample collection and laboratory analysis. Funding was provided by the Western Indian Ocean Marine Science Association (WIOMSA) grant no: MASMA/OP/2018/01.

**Conflicts of Interest:** The authors declare that the research was conducted in the absence of any commercial or financial relationships that could be construed as a potential conflict of interest.

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
