# Peer review of "Seasonal Occurrence and Relative Abundance of Marine Fish Larval Families over Healthy and Degraded Seagrass Beds in Coastal Kenya"

_diversity, doi:10.3390/d14090730_

Round 1
Reviewer 1 Report
The manuscript titled “Patterns of fish larval supply in seagrass habitats of Watamu and Diani in coastal Kenya” by Mwaluma et al. aims to identify patterns of fish larvae across two sites in Kenya. Identifying fish larval supply patterns in seagrass beds is of increasing importance as these nursery habitats are under threat due to anthropogenic stressors. The objectives of this study are to quantify larval supply to 1) describe seasonal patterns, 2) identify differences across two sites in Kenya, 3) relate these patterns to biotic and abiotic factors and 4) assess the effects of habitat fragmentation. While the objectives are clearly identified in the introduction, I am not convinced that the data is presented in a way that meets these objectives. The issues I raise may be influenced by a lack of clarity in the methods and results. Specific comments are explained below:
Line 69. Please include units.
Line 71. It is not clear how the healthy and degraded sites are defined. While there is an explanation to follow, it may be important to either provide empirical data from surveys or a citation that clearly shows this distinction between “healthy” and “degraded” sites.
Line 76. What caused this degradation at these two sites? I am not convinced that the sites are degraded versus simply just more patchy habitats.
Line 78. Replace with “3km from shore”.
Line 84. How often was temperature and salinity measured?
Line 90. How might these plankton tow methods affect the results (i.e., How would this be different from light traps, or another kind of more passive data collection?)
Line 92. What time of day were tows conducted?
Line 107. Is there a reason why different units are used for zoop and fish eggs versus fish larvae?
Line 114. Please make it clear that plankton abundance was tested in these models.
Linen 115. Where are the results from these tests? Might be useful to put a table in supplemental.
Line 123. What about biomass of planktivores? What other biotic factors were not accounted for here that may influence larval supply?
Line 130. Where are these data – are they presented graphically? This seems important to include if the differences between healthy and degraded sites matter.
Line 132. This only seems to be clear in 2019, but changes in later years, which is interesting.
Figure 2. This should be listed as Figure 1, correct?
Figure 2. Why is there no figure for DO?
Figure 2A. The NEM and SEM months seem to be off here.
Figure 2. Do the error bars show SE or SD? Please put this in the figure caption.
Line 171. What is the purpose of looking at site differences? Why not instead focus on healthy vs. degraded sites.
Line 172. Does zooplankton include fish larvae? Please make clearer.
Line 174. Also note that Diani was just more variable in general, whereas Watamu was more consistent but the pattern of degraded vs healthy was not clear at either site.
Figure 3. Please make the years clearer.
Figure 3. You can remove the decimal places on the Y axis since all are whole numbers.
Figure 3B. When I see this figure, I immediately want to know why fish eggs were in such higher abundance during NEM at degraded stations in Watamu.
Table 1. The table caption is not clear – this table seems to be showing total abundance values (not variation), correct?
Line 209. Please list the actual P value, not just > 0.05.
Figure 4. Why are these numbers different from Figure 3C? Are these the total abundances over the entire year, across all seasons? I think that could be OK if the effort was definitely the same across seasons, otherwise I would advocate for average abundance per sampling trip.
Line 215. Please combine this sentence with the following sentence (216) to make clearer.
Line 219. Scaridae does not look scarce to me.
Line 233. What about salinity?
Figure 5. Please list the environmental variables here, otherwise it is confusing with the sites listed after.
Line 251. Fish eggs and phytoplankton are not statistically significant – therefore it would be best to focus on Chl a and zooplankton.
Line 265. Would be useful to have a citation here.
Line 306. Were samples taken with regards to lunar cycle?
Line 317. One thing that jumps out to me is that degraded stations show just as high, if not higher diversity of species as ‘healthy’ stations. Why do you think this is?
Author Response
Reviewer No. 1
Comments and Suggestions for Authors
The manuscript titled “Patterns of fish larval supply in seagrass habitats of Watamu and Diani in coastal Kenya” by Mwaluma et al. aims to identify patterns of fish larvae across two sites in Kenya. Identifying fish larval supply patterns in seagrass beds is of increasing importance as these nursery habitats are under threat due to anthropogenic stressors.
The objectives of this study are to quantify larval supply to 1) describe seasonal patterns, 2) identify differences across two sites in Kenya, 3) relate these patterns to biotic and abiotic factors and 4) assess the effects of habitat fragmentation. While the objectives are clearly identified in the introduction, I am not convinced that the data is presented in a way that meets these objectives. The issues I raise may be influenced by a lack of clarity in the methods and results. Specific comments are explained below:
Line 69. Please include units.
Units have been indicated (see line 66)
Line 71. It is not clear how the healthy and degraded sites are defined. While there is an explanation to follow, it may be important to either provide empirical data from surveys or a citation that clearly shows this distinction between “healthy” and “degraded” sites.
Has been elaborated in text (see line 68-82)
Line 76. What caused this degradation at these two sites? I am not convinced that the sites are degraded versus simply just more patchy habitats.
Has been elaborated in text (see line 78-82)
Line 78. Replace with “3km from shore”.
Has been replaced in text (see line 67)
Line 84. How often was temperature and salinity measured?
Has been indicated in text (see line 87)
Line 90. How might these plankton tow methods affect the results (i.e., How would this be different from light traps, or another kind of more passive data collection?)
The plankton tows mainly capture pre-settlement larvae (planktonic), which obviously may bias results by underestimating other development stages of larvae particularly those with strong swimming abilities which tend to avoid nets. Light traps would be useful for collection post-settlement larvae but however with biases towards phototactic larvae. Future studies can be designed to utilise both methods in order to get a broader representation of taxa and families that settle within each habitat. (see line 340-350)
Line 92. What time of day were tows conducted?
The time of the day has been cited in the methods
Line 107. Is there a reason why different units are used for zooplankton and fish eggs versus fish larvae?
Different units are used for zooplankton, fish eggs and larvae due to scarcity of fish larvae in the samples. For sparse samples like fish larvae, density is multiplied by 100 to make it no.100-3 as compared to zooplankton which is calculated as no.m-3
Line 114. Please make it clear that plankton abundance was tested in these models.
Has been indicated in text (see line 153-156)
Linen 115. Where are the results from these tests? Might be useful to put a table in supplemental.
Supplementary table (Table S1) now included
Line 123. What about biomass of planktivores? What other biotic factors were not accounted for here that may influence larval supply?
Biomass of planktivores was not calculated in this study, but zooplankton was estimated as density (no.m-3) density. Other biotic factors that may influence larval supply were phytoplankton density which was not included in this paper as. The other biotic factors have been described in the text (see line 161)
Line 130. Where are these data – are they presented graphically? This seems important to include if the differences between healthy and degraded sites matter.
They are not represented graphically but included in supplementary Table S1
Line 132. This only seems to be clear in 2019, but changes in later years, which is interesting.
The data has been re-analyzed and Chlorophyl graphs re-drawn (See Figure 2B)
Figure 2. This should be listed as Figure 1, correct?
I have inserted a missing figure (map of study sites) labelled as Figure 1. So it correctly remains as Figure 2
Figure 2. Why is there no figure for DO?
The figure has been updated to include D.O (Figure 2 A)
Figure 2A. The NEM and SEM months seem to be off here.
The values have been re-confirmed and the graph re-drawn
Figure 2. Do the error bars show SE or SD? Please put this in the figure caption.
This has been inserted in figure caption (±SE)
Line 171. What is the purpose of looking at site differences? Why not instead focus on healthy vs. degraded sites.
The sentence is clarified to indicate healthy vs degraded sites (See line 192)
Line 172. Does zooplankton include fish larvae? Please make clearer.
It excludes fish larvae and it is explained in the methods (See line 137-140)
Line 174. Also note that Diani was just more variable in general, whereas Watamu was more consistent but the pattern of degraded vs healthy was not clear at either site.
We agree with the observation and have added in the text (See line 215-217)
Figure 3. Please make the years clearer.
Done
Figure 3. You can remove the decimal places on the Y axis since all are whole numbers.
This has been done
Figure 3B. When I see this figure, I immediately want to know why fish eggs were in such higher abundance during NEM at degraded stations in Watamu.
The high abundance of fish eggs during the NEM could have been caused by a spawning event of fish from family that have a preference for sandy open areas such as Bleniidae and Gobiidae as observed by Mwaluma et al 2011. We have added a statement in the discussion describing this (See line 315-319)
Table 1. The table caption is not clear – this table seems to be showing total abundance values (not variation), correct?
Table caption has been updated appropriately
Line 209. Please list the actual P value, not just > 0.05.
Actual P value has been inserted (See line 209)
Figure 4. Why are these numbers different from Figure 3C? Are these the total abundances over the entire year, across all seasons? I think that could be OK if the effort was definitely the same across seasons, otherwise I would advocate for average abundance per sampling trip.
Figure 3C shows seasonal abundance patterns while figure 4 shows total abundance for the entire year distributed in degraded and healthy habitats. Yes they were based on standardised sampling effort across seasons.
Line 215. Please combine this sentence with the following sentence (216) to make clearer.
The sentence has been combined (See line 221)
Line 219. Scaridae does not look scarce to me.
The sentence has been clarified to indicate the scarce families (see line 225)
Line 233. What about salinity?
A statement on the effect of salinity has been added (see line 253)
Figure 5. Please list the environmental variables here, otherwise it is confusing with the sites listed after.
Figure caption has been updated to include environmental variables
Line 251. Fish eggs and phytoplankton are not statistically significant – therefore it would be best to focus on Chl a and zooplankton.
The sentence has been reconstructed focusing only on Chl a and zooplankton (see line 262)
Line 265. Would be useful to have a citation here.
We have included the following citations -Daudi et al, 2010; Aboud et al 2017; Cowburn et al 2018) (see line 272-278)
Line 306. Were samples taken with regards to lunar cycle?
No. however we have included this as a recommendation for future assessments.
Line 317. One thing that jumps out to me is that degraded stations show just as high, if not higher diversity of species as ‘healthy’ stations. Why do you think this is?
We have improved on the explanation and included your perspective. Thank you.

Author Response
Reviewer no. 2
This is an important study in a poorly studied global coastal area and covering fish early life histories that are seldom investigated by marine ichthyologists (in comparison to the juvenile and adult life stages). Not only does it document fish larval diversity and relative abundance in association with coastal seagrass beds in two lagoons, it also provides insight that healthy seagrass habitats are more likely to have higher larval densities than degraded ones.
The following issues require the attention of the authors;
Lines 2-3: Suggested revised title - Fish larval diversity and relative abundance in selected seagrass habitats of two shallow lagoons on the Kenya coast
The authors appreciate the suggested revised title, however since the study did not assess diversity, the prefer to remain with the current title
Line 43: I agree on the nursery habitat provided by seagrass beds but am not so sure about spawning grounds for commercially important fish species. Most of the species that are likely to spawn in the seagrass beds belong to non-commercially important taxa.
The sentence has been reconstructed to exclude spawning grounds
Line 56: Reword “…meadows, thus limiting their nursery functions…”
Line 56 has been reworded
Line 60: Insert “seagrass” before “fragmentation”
“Seagrass” has been inserted before “fragmentation” in Line 60
Lines 61: The main thrust of this study was directed towards seasonal and habitat degradation influences on the associated larval fish assemblages. However, there was also a major difference between the type of seagrass in the two study areas – Watamu comprised mixed seagrass beds of pioneer species whereas Diani comprised climax seagrass beds dominated by a single species. Clearly there needs to be a focus on the similarities/contrasts between fish assemblages in these seagrass habitat types? Thank you for your comment indeed this is true. These differences can be discerned from Table 1 which compared seasonal larval assemblages from the two sites
Lines 83-88: No mention is made under the environmental variables of pH measurements. Measurements for PH were done using the YSI multiparameter probe alongside temperature and salinity. This has been added in the text in line 136
Why was pH even considered since seawater is a major buffer in terms of pH and therefore leads to only minor variations? Variations in pH were found to significantly differ between seasons, therefore the factor has been retained in the results
Line 91: What was the diameter of the mouth of the plankton net?
The net diameter was 0.5m. This has been inserted in line 91
Line 92: Were the samples collected from the surface, middle or bottom of the water column?
The samples were collected from the surface layer of the water column as now indicated line 142
Line 93: At what time were the samples collected and were diurnal and nocturnal plankton samples ever collected and compared? What was the bias in larval composition resulting from only diurnal samples being collected? Samples were collected between 0900hrs and 1400 hrs against the oncoming high tide at a constant speed of between 0.5-1.8 knots (see line 144). Nocturnal samples were not collected.
Pages 3, 4 and 5: No mention of pH – so why is it used in Figure 5?
Fig. 2 has been amended to include pH. The same has been included in the methods (see line 136) and description of results (see lines 198-200)
Lines 243-245: Suddenly there is mention of pH but no actual values are given. Values have been presented graphically in Fig. 2, and now have been mentioned in results section (198-200). Can the authors link the actual pH values measured in this study to fish larval studies that have focused on the influence of this environmental variable to fish larval health/growth/development? If not, remove this variable from this study. We considered pH as an important parameter because studies have shown that reduction of pH levels can affect growth rates of fish larvae. Further to this, our findings show significant difference between seasons, due to this, the factor was retained in the results (see line 180-182)
Line 256: Previous findings from where?
Along the Kenyan coast. This has been added to the line 268
Lines 259-260: If there is water above the seagrass beds, could this be termed ‘open water’? This water Presumably the plankton net was pulled through ‘open water’ when sampling – otherwise the net would be clogged with seagrass fronds. Thus sampling during this study was not ‘in’ the seagrass beds sensu stricto.
Open waters in this case refers to oceanic waters beyond the fringing reef which are periodically flushed into the bay during high tides. Sampling was carried out only during high tides above the seagrass beds.
Lines 260-268: Comments on spawning locality can only be made if the eggs (not larvae) were effectively sampled and identified. A 500 um mesh net will not sample fish eggs of many species effectively and eggs that were collected were not identified in this study. I therefore do not understand how statements on spawning locality can be made with any certainty. The authors should therefore restrict their comments to the occurrence of fish larvae only. This comment is appreciated and noted. Statements on spawning locality have been corrected and replaced with occurrence of fish larvae. See 269-270
It is also unlikely that phytoplankton (chlorophyll a) directly influence fish larval abundance – unless it can be shown that these larvae are consuming phytoplankton. It is more likely that zooplankton will influence fish larval abundance since this food source has been shown to be very important to fish larvae. The relationship between phytoplankton concentrations and zooplankton abundance is seldom linear. This is noted and appropriately corrected in the sentence (see line 293-295)
I also question the value of seagrass beds to drifting preflexion larvae that have poor swimming abilities. Only at the postflexion larval and early juvenile stages will seagrass beds play a role, i.e. it is the early juveniles (not early larvae) that make use of seagrass beds for food and shelter from predators. Early juveniles would not be effectively sampled by the plankton gear used in this study due to their net avoidance capabilities and occupation of the bottom seagrass beds (i.e. they are demersal/epibenthic and not pelagic where the bongo net was operated). This is noted and elaborated in the conlusion (see line 339-344).
Line 277 and 282: A study cannot “observe” (see) – a study can have findings.
The word “observe” has been replaced with “found” See line 302 and 307
Lines 285-287: Temperature has an influence on dissolved oxygen retention and season influences temperature levels. So what is the primary physico-chemical driver of fish larval abundance in these sampling localities?
The primary physico-chemical driver would be temperature. This has been rephrased see line 310
Lines 290-291: Why would one expect a ‘match’ between fish egg density and phytoplankton density since neither is dependent on the other? Fish egg density may also not match fish larval density due to the eggs having hatched or, alternatively, the eggs sampled may be different species to the species of fish larvae sampled. This study did not determine the species identity of either the eggs or the larvae – so assumptions or conjectures in this match/mismatch concept cannot be made. Assumptions or conjectures in match/mismatch concept has been deleted
Lines 294-296: This sentence is incomplete and requires rewording. In addition, why is larval ‘survival’ of a particular species dependent on what happens during the adult life stages of the same species?
Sentence has been reworded. I interpret the sentence to mean that fish larvae (at all stages) will rely on favourable biotic and abiotic factors for successful recruitment (see line 305-306).
Lines 296-297: Elaboration is required to justify why these drivers of fish egg distribution only apply to degraded sites and not natural sites.
This elaboration has been provided see lines 317
Lines 298-309: This paragraph requires considerable expansion and a separation into at least three paragraphs, all of which need to emphasize the limitations of the sampling gear in capturing late larvae and early juveniles of species recruiting into the seagrass nursery areas.
The paragraph has been considerably expanded to include the limitations of the sampling gears as suggested see line 339-349
Firstly, it needs to be acknowledged that small, benthic/demersal species (e.g. Blenniidae and Gobiidae) do not have mass migrations to offshore or pelagic spawning grounds – they are likely to spawn locally and therefore the larvae will be well represented in the plankton samples collected during the current study. Secondly, large commercially important species belonging to families such the Lethrinidae and Lutjanidae are likely to spawn in deeper offshore coastal waters, with pelagic dispersal of larvae meaning that these species are likely to be poorly represented in plankton samples in and around the shallow-water seagrass beds. However, their postlarvae/early juveniles, which will not be properly sampled by a bongo plankton net, are likely to be well represented in the seagrass beds. Thirdly, the authors need to explain why the Engraulidae, which are usually pelagic coastal water spawners were so well represented in the seagrass bed samples, and the Syngnathidae which are seagrass bed residents, had higher larval densities in bare areas rather than over the seagrass beds.
I also suggest a conceptual diagram be created to illustrate the distributional patterns of larvae belonging to the different fish families at different times of the year. This will help readers to visually understand what is happening from a fish larval perspective.
It is impossible to include a conceptual diagram because temporal variables are stochastic and therefore difficult to predict the distribution of larvae at any given time.
Somewhere in the discussion there needs to be an emphasis that the plankton sites at the Watamu and Diani localities are not comparable on at least two scores, firstly the geographic separation and secondly the different types of seagrass habitat in the two areas. Furthermore, this new paragraph needs to be explicit as to whether the decline in fish larval densities in degraded seagrass habitat within the above two localities is the same, similar or different.
This has been taken into context and added in the discussions

Round 2
Reviewer 1 Report
Thank you for your responses and revisions. I believe that the changes that have been made have improved the manuscript. My only remaining comments are regarding Figure 2:
1. Please check the X-axis of Figure 2A – it appears as though months within NEM and SEM differ across the years.
2. Please fix the labeling of the figures within Figure 2 (A-E).
Author Response
Comments and Suggestions for Authors
Thank you for your comments and second revision. I have concluded the second review as follows;
- Please check the X-axis of Figure 2A – it appears as though months within NEM and SEM differ across the years. We have rectified by deleting a conflicting figure which initially appeared as Figure 2A.
- Please fix the labeling of the figures within Figure 2 (A-E). This has been done
Reviewer 2 Report
I am satisfied with the responses of the authors to my comments and queries but am still unhappy with the title. Fish "larval supply" was not measured by this study - the relative abundance of fish larval families (not species) associated with the seagrass beds was measured. There is also no mention of the two major aspects of the study - namely the seasonal occurrence and relative abundance of the larvae over healthy and degraded seagrass beds. I therefore recommend a rewording of the title along the following lines (or something similar) "Seasonal occurrence and relative abundance of marine fish larval families over healthy and degraded seagrass beds at two localities on the Kenya coast".
Author Response
I am satisfied with the responses of the authors to my comments and queries but am still unhappy with the title. Fish "larval supply" was not measured by this study - the relative abundance of fish larval families (not species) associated with the seagrass beds was measured. There is also no mention of the two major aspects of the study - namely the seasonal occurrence and relative abundance of the larvae over healthy and degraded seagrass beds. I therefore recommend a rewording of the title along the following lines (or something similar) "Seasonal occurrence and relative abundance of marine fish larval families over healthy and degraded seagrass beds at two localities on the Kenya coast". We thank the reviewer for this suggestion. We have adopted the new proposed title ""Seasonal occurrence and relative abundance of marine fish larval families over healthy and degraded seagrass beds at two localities on the Kenya coast".